# Exploring the Effect of Probiotics, Prebiotics, and Postbiotics in Strengthening Immune Activity in the Elderly

**DOI:** 10.3390/vaccines9020136

**Published:** 2021-02-08

**Authors:** Hiroyasu Akatsu

**Affiliations:** Department of Community-Based Medical Education, Graduate School of Medical Sciences, Nagoya City University, Nagoya 467-8601, Japan; akatu@med.nagoya-cu.ac.jp; Tel.: +81-52-851-5511

**Keywords:** probiotic, prebiotic, postbiotic, elderly, gut microbiota (GM), influenza, vaccination, immunity

## Abstract

Vaccination is the easiest way to stimulate the immune system to confer protection from disease. However, the inefficacy of vaccination in the elderly, especially those under nutritional control such as individuals receiving artificial nutrition after cerebral infarction or during dementia, has led to the search for an adjuvant to augment the acquired immune response in this population. The cross-talk between the gut microbiota and the host immune system is gaining attention as a potential adjuvant for vaccines. Probiotics, prebiotics, and postbiotics, which are commonly used to modulate gut health, may enhance the immune response and the effectiveness of vaccination in the elderly. This review summarizes the use of these gut modulators as adjuvants to boost both the innate and acquired immune responses in the elderly under nutritional control. Although the clinical evidence on this topic is limited and the initial findings await clarification through future studies with large sample sizes and proper study designs, they highlight the necessity for additional research in this field, especially in light of the ongoing COVID-19 pandemic, which is disproportionately affecting the elderly.

## 1. Introduction

The world is currently experiencing four “megatrends” in global demographics: Population aging, urbanization, climate change, and globalization. Driven by advances in public health, medicine, and economic and social development, human life expectancy has been increasing. Globally, there are currently 727 million people aged ≥65 years, and they constitute 9.3% of the total population. This proportion is expected to increase continuously to 16% over the next three decades, when one in six people worldwide will be aged ≥65 years old (Figure 1a) [1]. Notably, this phenomenon is particularly true of Japan’s super-aging society. It is estimated that the proportion of elderly individuals (>65 years old) will increase to over 30% in the near future [2].

Unfortunately, immunosenescence is one consequence of a prolonged lifespan, and elderly people are therefore more vulnerable to infectious diseases. Strikingly, thymus atrophy begins as early as puberty, and continues at a rate of about 1% per year. Consequently, fewer naïve T cells are produced with time, and T-cell receptor variants become unable to diversify and respond to newly encountered antigens. Similarly, the ability of B cells to produce immunoglobulin also declines with age, as do the activities of neutrophils and natural killer (NK) cells [3,4,5,6,7,8,9,10]. Thus, the elderly is disproportionately affected by infectious diseases. For instance, seasonal influenza causes substantial morbidity and mortality among the elderly, and this imposes a financial burden worldwide (Figure 1b) [11,12,13,14]. As the new global pandemic of coronavirus (COVID-19) surged in 2020, the elderly were at a significantly greater risk of infection, and had a significantly higher mortality rate, than any other age group, as reported by the World Health Organization [10,15]. 

Vaccination is one of the most effective strategies to protect the elderly against common infectious diseases. However, the efficacy of vaccination in this group is limited by immunosenescence. For example, the clinical effectiveness of the influenza vaccine decreases with age, from 70–90% efficacy in young, healthy adults to 17–53% efficacy in the elderly [16]. Moreover, the immune functions of elderly people who receive artificial nutrition after cerebral infarction or during dementia are expected to deteriorate even further. Therefore, the finding that influenza vaccination is improved by interventions that stimulate the gut flora is extremely important. Various biotic interventions directed towards the intestinal flora in the elderly are expected to improve their immune status and the effects of vaccination. Several studies have reported that the consumption of probiotics modulates immune responses to the influenza vaccine, providing insights into the potential adjuvant effects of probiotics on vaccination.

In this review, we summarize the results of a series of intervention studies on the intestinal immunity in elderly subjects under constant nutritional control. The research subjects in these studies had been institutionalized and bed-ridden for a long period of time under specific artificial nutritional management, mainly managed by research groups in Japan. The results of these studies suggest that modulation of the gut microbiota (GM) substantially affects the efficacy of vaccination.

## 2. Interplay between Gut Microbiota and Immunity in the Elderly

The relationship between the GM and the host defense system is the principle underlying Mechnikov’s yogurt longevity hypothesis [17]. The composition of the GM differs between individuals and has emerged as an important immune modulator, closely associated with an individual’s health, including the risk of disease development. The cross-talk between the GM and the host’s immune system is achieved through molecular interactions after bioactive metabolites (short-chain fatty acids) are produced or through interactions with the host’s immune cells through cell-surface molecules (peptidoglycans and lipopolysaccharides) [10,18,19,20]. Notably, recent studies have reported the effects of the GM on the response to vaccination, including its stimulation of specific CD8^+^-cell differentiation, B-lymphocyte growth and differentiation, and production of specific immunoglobulin A molecules (IgAs) [21,22,23,24]. For example, Toll-like receptor 5 (TLR5) mediates the sensing of common GM-produced compounds by immune cells, and bacterial flagellin promotes host antibody titers and plasma cell growth [25].

The composition of the GM is continuously undergoing massive changes, which are dependent upon the diet, health status, drug intake, lifestyle, and age of the host. Therefore, the composition of the GM in the elderly differs markedly from that in the younger population, with a lower prevalence of *Bifidobacterium* and a higher prevalence of potentially dangerous bacteria, such as *Clostridium* and enterobacteria [26,27]. When this is coupled with immunosenescence, the effectiveness of vaccination in the elderly population can be as low as 20%. Therefore, researchers are continually seeking various interventions to improve the effectiveness of vaccination in the elderly [28,29].

Akatsu et al. (2011) reported differences in both the effectiveness of influenza vaccination and the intestinal microbial groups between adult individuals and the elderly [30]. Body mass indices and blood albumin levels decreased in the sequence: Healthy adults ≒ healthy elderly > elderly with enteral nutrition (EN) > elderly with total parenteral nutrition (TPN). At six weeks post-vaccination, the antibody titers for B antigens were significantly higher in the healthy adults than in the three groups of elderly subjects. However, there was no significant difference in the anti-H1N1 and -H3N2 antibody titers among the four groups tested. Interestingly, a fecal microbiota analysis indicated a significant drop in the total *Bifidobacterium* counts in the elderly groups (10^6^ colony-forming units [CFU]/g) compared with that in the healthy adult group (10^10^ CFU/g). A similar trend was observed in the gut occupancy rate of *Bifidobacterium*, which dropped from 4% to 0.5% in the following order: Healthy adults ≒ healthy elderly > EN elderly > TPN elderly. These outcomes suggest that age, nutritional status, and GM all affect the efficacy of vaccination.

## 3. Interventions in the Gut Environment to Enhance Immunity in the Elderly

Here, we analyzed existing systematic reviews together with various interventional studies [31,32,33,34]. The use of probiotics and prebiotics to enhance vaccine effectiveness has emerged as a feasible and attractive strategy. Probiotics are live microorganisms that, when taken in sufficient amounts, provide health benefits to the host [35]. Some may even act as part of the intestinal microflora. Studies have reported the ability of probiotics to induce cellular immunity by promoting phagocytes and NK cells, to enhance the effects of vaccines, to promote IgA secretion, and to ameliorate the incidence and duration of infections in the elderly [36,37,38,39,40,41,42,43,44].

In contrast, prebiotics are substances that are selectively used by the microorganisms that live in our bodies and that positively affect our health [45]. The administration of prebiotics promotes the growth of *Bifidobacterium* in the gut, modulates the B-cell response, and enhances the Th1-dependent immune responses, NK cell activity, and interferon-γ (IFN-γ) production, which in turn augment the effects of vaccines [46,47,48,49,50,51].

Similar to probiotics and prebiotics, postbiotics have recently been tentatively defined as bioactive compounds produced during a fermentation process (including inactive microbial cells, cell constituents, and metabolites) that support health and/or well-being [52]. Postbiotics, particularly heat-killed bacteria, have been widely used in various applications, including foods, cosmetics, and pharmaceuticals. Although the exact mechanism of their action is still not fully understood, their immune-modulating effects are undeniable [52]. Li et al. (2001) reported the ability of heat-killed *Lactobacillus rhamnosus* GG, when consumed, to suppress proinflammatory mediators and enhance the activity of anti-inflammatory mediators in the liver, plasma, and lung [53]. Similarly, heat-killed bacteria exert immunoregulatory effects that are as effective as those exerted by live bacteria, activate splenocytes, and dendritic cells, and induce the Th1-immune response and the production of tumor necrosis factor α (TNF-α), interleukin 6 (IL-6), and IL-10 [26,53,54]. Studies have demonstrated that they affect the immune system by raising mucin levels and promoting the development of claudin and occludin, confirming their potential as possible functional ingredients [55]. 

Although the immune-response-promoting functions of these beneficial microbes and their byproducts have been widely reported, their effectiveness as adjuvants in vaccination, especially for the elderly, remains unconfirmed. Only limited studies with small sample sizes are available as references. This review highlights the potential effects of probiotics, prebiotics, and postbiotics in enhancing the immune defenses, and the effectiveness of anti-influenza vaccines in elderly subjects under nutritional control (Table 1). In this review, only some of the studies will be discussed.

### 3.1. Effect of Probiotic Bifidobacterium Strain on the Immune Activity of Elderly EN Subjects

Several interventional studies have demonstrated that the administration of probiotics tends to enhance vaccine effects in the elderly, including those under nutritional control [38,39,56,57,58]. In a single-center, double-blind, placebo-controlled, parallel-group study by Akatsu et al. (2013) [56], 45 elderly patients receiving enteral tube feeding, with a mean age of 81.7 years, were divided into two groups that received either a placebo (2 g of dextrin; *n* = 22) or Bifidobacterium longum BB536 powder (5 × 10^10^ CFU/2 g; *n* = 23) twice a day for 12 weeks. An influenza vaccine (types A/H1N1, A/H3N2, or B) was introduced four weeks after the intervention. Notably, there were significant increases in the total bifidobacteria (*p* < 0.05 vs. placebo), B. longum subsp longum (*p* < 0.01 vs. placebo), and B. longum strain BB536 (*p* < 0.01 vs. placebo) counts. The rate of elevated fever was lowest (*p* < 0.01 vs. placebo) and the bowel movement rate was highest in the BB536 group (*p* < 0.01 vs. placebo) [56]. These data indicate that the ingestion of strain BB536 improved the health of elderly patients, who commonly experience difficulty with defecation and low-abundance Bifidobacterium in their GM [25]. Unfortunately, no intergroup differences were detected in the subjects’ antibody titers. The ingestion of BB536 significantly improved the anti-H1N1 antibody titers (week 6, *p* < 0.05) in elderly patients, to antibody titers ≥ 20. The ingestion of BB536 also tended to increase the serum IgA levels (week 6, *p* = 0.09; week 16, *p* = 0.07) compared with those in the placebo group. The NK cell activities in the placebo group tended to decline, whereas they tended to remain stable in the BB536 group. Interestingly, BB536 markedly optimized the innate immune functions of immunosuppressed elderly subjects, evident as significantly higher NK cell activities in those patients with previously low NK cell activities (≤55%), in weeks 8 (*p* < 0.05) and 12 (*p* < 0.05) after administration.

Another interventional study of the same strain (B. longum BB536) by Namba et al. (2010) [39] suggested that the long-term administration of BB536 improved the innate immunity of the elderly and reduced their risk of influenza and fever. Twenty-seven elderly subjects, who were receiving enteral tube feeding, with a mean age of 86.7 years were recruited and administered BB536-containing foods (1 × 10^11^ CFU) for 5 weeks (Phase 1). In week 3, an influenza vaccine (type A/H1N1, A/H3N2, or B) was given to all the elderly subjects. In Phase 2, the subjects were randomized into two groups, receiving either BB536 (*n* = 13) or a placebo (dextrin; *n* = 14) for the next 14 weeks. Notably, the consumption of BB536 resulted in significantly lower incidences of influenza (*p* = 0.041) and fever (*p* = 0.046) compared with the placebo group. However, the consumption of BB536 did not help to maintain the antibody titers against the influenza vaccines. In the Phase 1 period, the consumption of BB536 significantly elevated the phagocytic and bactericidal activities of neutrophils (*p* < 0.01) and the activities of NK cells (*p* < 0.01) in the elderly subjects. An increase in NK cell activities is essential in the defense against viral infections. NK cells are the major source of IFN-γ, a potent antiviral immunostimulatory cytokine in humans [63,64]. However, these effects were gradually lost in the placebo group (week 10 vs. week 20, *p* < 0.01) in the Phase 2 period, whereas the continued consumption of BB536 helped to maintain higher levels of the phagocytic and bactericidal activities of neutrophils and the activities of NK cells throughout the Phase 2 period.

Taken together, these clinical findings support the notion that the prolonged ingestion of BB536 is a potential prophylactic approach to improving the innate immunity of elderly patients. However, clinical data on its effects on acquired immunity are still limited and await future large-scale studies.

In addition, some studies have been performed on the immune activity of elderly by Lactobacillus strains [56,57]. Bosch et al. (2012) demonstrated that three-month supplementation of Lactobacillus plantarum CECT 7315/7316 significantly improved influenza-specific IgA level [58].

### 3.2. Effect of Heat-Killed Lactobacillus Strains on Immune Activity in Elderly with Oral Intake

The oral intake of heat-killed *Lactobacillus pentosus* b240 confers protection against the common cold in the elderly [37]. Therefore, the potential utility of heat-killed probiotics, also known as postbiotics, is gaining attention to improve the immunity of the elderly. However, only limited studies of postbiotics have shown that they enhance the efficacy of vaccines in the elderly [59,60]. In a study by Akatsu et al. (2013) [59], 15 elderly volunteers from a nursing home were recruited and randomly divided into two groups, receiving either jelly containing 10^10^ heat-killed *Lactobacillus paracasei* MoLac-1 (MoLac group, *n* = 8) or jelly without heat-killed lactobacilli (placebo group, *n* = 7) for 12 weeks. All the participants were given same providing a meal three times every day and an influenza vaccine (type A/H1N1, A/H3N2, or B) three weeks after the intervention. Unfortunately, no differences were detected in the serum Ig (IgA, IgM, or IgG) levels, NK cell activities, or bactericidal or phagocytic activities of neutrophils between the MoLac and placebo groups throughout the study. However, the ingestion of *L. paracasei* MoLac-1 significantly improved the hemagglutination inhibition (HI) titers against the influenza antigens (type A/H1N1, *p* < 0.05; A/H3N2, *p* < 0.01; B, *p* < 0.05). Furthermore, *L. paracasei* MoLac-1 tended to produce higher HI titers than the placebo group (*p* = 0.09). Although the ingestion of *L. paracasei* MoLac-1 tended to improve the effects of these vaccines in the elderly, no effect was observed on other immunological parameters. The small sample size limited the accuracy and precision of the study outcomes.

Another study by Maruyama et al. (2016) [60] evaluated the effects of another strain of heat-killed bacteria, L. paracasei MCC1849, on the immune functions and vaccine efficacy in the elderly. Forty-two elderly subjects, aged ≥65 years, were recruited from two nursing homes and assigned to the LP group (ingesting jelly containing 10^10^ heat-killed L. paracasei MCC1849; *n* = 21) or the placebo group (jelly without lactobacilli; *n* = 21). Three weeks after the jelly was consumed, all the subjects were administered an influenza vaccine (type A/H1N1, A/H3N2, or B). Consistent with the previous study [59], no significant differences in immune parameters were observed between the groups throughout the six weeks of the interventional study. Interestingly, the antibody responses to type A/H1N1 and B antigens were significantly improved (*p* < 0.05) in the oldest participant subgroup (aged ≥85 years; *n* = 11) of the LP group. This outcome suggests that heat-killed L. paracasei MCC1849 enhanced the immune functions of the oldest subjects, who likely suffered the greatest immunosenescence.

The use of postbiotics is attracting attention because the administration of dead or inactivated cells reduces the risks associated with the administration of live bacteria, especially in immunocompromised individuals, such as the elderly. However, further large-scale studies are required to fully understand the effects of postbiotics in promoting the immune functions of this group.

### 3.3. Effect of Prebiotics on the Immune Activity of Elderly EN Subjects

Past studies have demonstrated the ability of probiotics to augment the immune functions of elderly subjects administered an influenza vaccine [38,39,56,57,58]. For instance, Vos et al. (2007) noted an increase in fecal *Lactobacillus* and *Bifidobacterium* counts after the ingestion of prebiotic galactooligosaccharides (GOS) in an influenza-vaccinated murine model [50]. These findings suggest that prebiotics boost the abundances of *Lactobacillus* and *Bifidobacterium* in the gut, with immunoregulatory effects on the host. However, only limited studies have addressed the immunoregulatory effects of prebiotics in the elderly [51,61,62,65,66].

In a study by Akatsu et al. (2016) [61], 23 bed-ridden elderly patients receiving percutaneous endoscopic gastrostomy were recruited for a 10-week intervention. The participants were randomized into two groups: Group F (*n* = 12) received a standard enteral formula (Fibren YH, from Meiji), together with 4.0 g of GOS and 0.4 g of bifidogenic growth stimulator (BGS) per day; Group C (*n* = 11), the control group, received Meibalance, which contains almost the same nutrients as Fibren YH, but without heat-treated lactic acid bacteria-fermented milk products or GOS and BGS supplementation. An influenza vaccine (type A/H1N1, A/H3N2, or B) was administered at week 4. Notably, the administration of GOS and BGS in Group F did not increase the number of *Lactobacillus* or *Bifidobacterium* bacteria, contrary to expectation. However, the number of *Bacteroides* bacteria, a species that has been reported to exert immunomodulatory effects [67,68], was significantly increased (*p* < 0.05) by GOS and BGS. Unlike Group C, Group F sustained high anti-H1N1 and -H3N2 antibody titers throughout the intervention, with a significantly higher seroprotective rate (64%; *p* < 0.05) against H3N2 compared with Group C (10%). However, the bias in the serum nutritional indices, such as the total protein and albumin levels, in Group F led to uncertainty in the ability of the prebiotics to modulate vaccine efficacy, the immune responses, and the GM in this intervention.

In an open-label, randomized, controlled trial, Nagafuchi et al. (2015) [62] used a study formula that ensured the same nutritional value for both the experimental group (Group F, supplemented with GOS, BGS, and pasteurized fermented milk products; *n* = 12) and the control group (Group C, standard enteral formula milk without prebiotics or fermented milk products; *n* = 12). In Group F, the *Bifidobacterium* count gradually increased throughout the intervention and was significantly higher (*p* < 0.05) than that in the control group after week 8. Unfortunately, although the seroprotective antibody titers against A/H1N1 were significantly augmented (*p* < 0.05) in both Groups C and F, the seroprotective antibody titers against the antigens did not differ between the two groups. However, the seroprotective antibody titers against the A/H1N1 antigens tend to be higher in Group F than in the control group at week 8. Interestingly, the seroprotective antibody titers against B antigen were significantly higher (*p* < 0.05) in Group C than in Group F at weeks 6 and 8.

These studies imply that the administration of prebiotics may maintain the antibody titers against influenza antigens in elderly subjects for an extended period; however, because the number of studies is limited, the effects of prebiotics on enhancing immune functions and vaccination efficacy are still inconclusive. Similar to the above observations, Bunout et al. (2002) [51] reported that administration of a prebiotic mixture (70% raftilose + 30% raftiline) or placebo significantly increased anti-Influenza B antibody titer in both prebiotic (*p* < 0.01) and placebo group (*p* < 0.01) compared to baseline, but without inter-group difference.

## 4. Conclusions

The interplay between the GM and the host immune system has greatly affected modern therapeutic interventions, and manipulating the GM to enhance the acquired immune response in the elderly is attracting interest. Mounting evidence suggests that altering the GM with probiotics, prebiotics, or postbiotics is a feasible way to enhance the effects of vaccination in the elderly. Although the currently available evidence is not robust, the use of probiotics, prebiotics, or postbiotics has tended to improve the immune responses of elderly subjects, including sustainable NK-cell activities and antibody titers, and to restore the GM balance (Figure 2). However, further research conducted using well-designed randomized trials with larger sample sizes is required to provide conclusive evidence of the ability of probiotics, prebiotics, and postbiotics to enhance the immune defenses and the effectiveness of influenza vaccination in the elderly under nutritional control.

## Figures and Tables

**Figure 1 vaccines-09-00136-f001:**
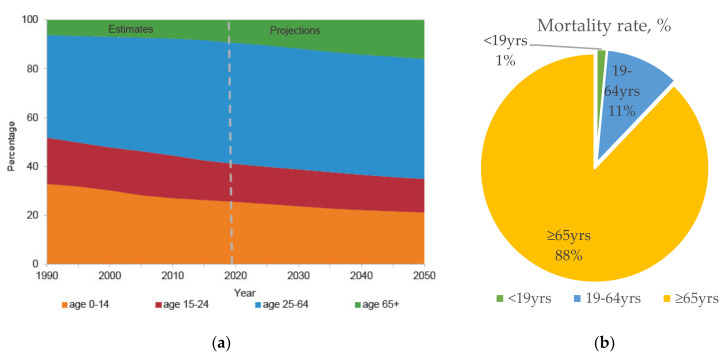
(**a**) Predicted global population trend by age group from 1990 to 2050 *. (**b**) Estimated number of annual influenza-associated deaths with underlying respiratory or circulation failure by age group, in the United States from 1976–2007 [12]. United Nations Department of Economic and Social Affairs, Population Division (2019) [2].

**Figure 2 vaccines-09-00136-f002:**
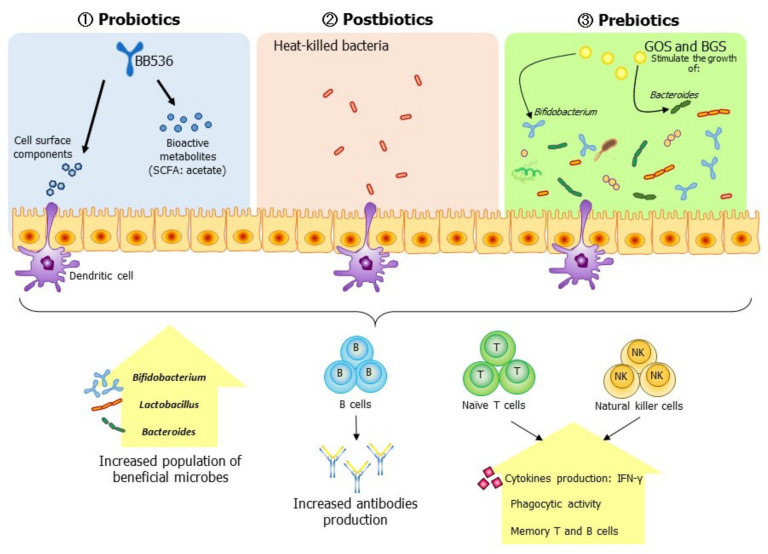
Modulation of the gut environment with probiotics, postbiotics, or prebiotics enhances the innate and acquired immunity of the elderly. Specifically, the ingestion of probiotics increases the population of *Bifidobacterium* in the elderly, augments the innate immune response by increasing the activities of natural killer (NK) cells, and increases the serum antibody titers against influenza antigen through cell interactions and the secretion of bioactive metabolites, such as short-chain fatty acid (SCFA), by immune cells. In contrast, postbiotics augment a vaccine’s effect by promoting the activation and secretion of serum antibodies against influenza antigens in the elderly. Prebiotics act as stimulants that promote the growth of the gut’s beneficial microbes (*Bifidobacterium, Lactobacillus*, and *Bacteroides*), thereby activating the host immune response via cellular interactions and bioactive metabolites from the beneficial microbes, thus increasing antibody production and innate immune responses to influenza. These points highlight the potential of an adjuvant effect but await confirmation by future research.

**Table 1 vaccines-09-00136-t001:** Effect of probiotics, prebiotics, and postbiotics on the immune responses of the elderly.

Reference	Type of Study	Subjects	Intervention	Period	Influenza Vaccination	Effect on Vaccination	Other Outcomes
Akatsu et al. (2013) [56]	Randomized, double-blind, placebo-controlled study	45 elderly patients aged 65 years and older	*Bifidobacterium longum* BB536 powder (5 × 10^10^ CFU/2 g, twice/day; *n* = 23) vs. placebo (*n* = 22).	12 w	At week 4 (A/H1N1, A/H3N2, and B)	A/H1N1 antibody titers ≥ 20 significantly increased in the probiotic group (*p* < 0.05) than the placebo group.	Tended to stimulate NK cells activity (*p* < 0.1); Tended to increase IgA levels (*p* < 0.1).
Namba et al. (2010) [39]	Phase I: Single arm	27 elderly residents aged 65 years and older	*Bifidobacterium longum* BB536 powder (1 × 10^11^ CFU/2 g/day; *n* = 13) vs. placebo (*n* = 14).	5w	At week 3 (A/H1N1, A/H3N2, and B)	No intergroup difference.	Reduce influenza and fever cases (*p* = 0.041); Stimulated NK cell activity and the neutrophils phagocytic and bactericidal activities.
	Phase II: Randomized, double-blind, placebo-controlled study			14 w			
Van Puyenbroeck et al. (2012) [57]	Randomized, double-blind, placebo-controlled trial	737 healthy elderly aged 65 years and older	*Lactobacillus casei* Shirota fermented milk (1.3 × 10^10^ bacteria/day; *n* = 375) vs. placebo (*n* = 362)	176 days	At day 21 (A/H1N1, A/H3N2, and B)	No intergroup difference.	-
Bosch et al. (2012) [58]	Randomized, double-blind, placebo-controlled trial	60 instituitionalized elderly aged 65 years and older	High-dose *Lactobacillus plantarum* CECT 7315/7316 (5 × 10^9^ CFU/20 g/day; *n* = 19) vs. low-dose *Lactobacillus plantarum* CECT 7315/7316 (5 × 10^8^ CFU/20 g/day; *n* = 14) vs. placebo (*n* = 15)	3 months	3 months before intervention (A/H1N1, A/H3N2, and B)	Significantly improved influenza-specific IgG level in high-dose group (*p* = 0.023). Significantly improved influenza-specific IgA level in high-dose group (*p* = 0.008) and low-dose group (*p* = 0.039).	Tended to improved influenza-specific IgM level in high-dose group (*p* = 0.054).
Akatsu et al. (2013) [59]	Randomized, double-blind, placebo-controlled study	15 elderly patients aged 65 years and older	Jelly containing heat-killed *Lactobacillus paracasei* MoLac-1 (1 × 10^10^ cells/day; *n* = 8) vs. placebo (*n* = 7).	12 w	At week 3 (A/H1N1, A/H3N2, and B)	Significantly improved antibody titers of A/H1N1 (*p* < 0.05), A/H3N2 (*p* < 0.01), and B (*p* < 0.05) in MoLac-1 group.	No significant difference in other immune parameters between groups.
Maruyama et al. (2016) [60]	Randomized, double-blind, placebo-controlled study	45 elderly patients aged 65 years and older	Jelly containing heat-killed *Lactobacillus paracasei* MCC1849 (LP; 1 × 10^10^ cells/day; *n* = 21) vs. placebo (*n* = 21).	12 w	At week 3 (A/H1N1, A/H3N2, and B)	The antibody responses to type A/H1N1 and B antigens were significantly improved (*p* < 0.05) in the oldest old subgroup (aged ≥ 85 years; *n* = 11) of the LP group compared with the placebo group.	No significant difference in other immune parameters between groups.
Akatsu et al. (2016) [61]	Randomized, double-blind, placebo-controlled study	23 elderly patients received percutaneous endoscopic gastrostomy	Enteral formula supplemented with GOS, bifidogenic growth stimulator (BGS) and pasteurized fermented milk products (*n* = 12) vs. control enteral formula (*n* = 11).	14 w	At week 4 (A/H1N1, A/H3N2, and B)	Test formula led to a high level of anti-H1N1 and H3N2 antibody titers throughout the intervention, with a significantly higher seroprotective rate (64%; *p* < 0.05) against H3N2 than in Group-C (10%)	No significant difference in other parameters between groups.
Nagafuchi et al. (2013) [62]	Open-label, randomized, controlled trial	24 elderly patients received percutaneous endoscopic gastrostomy	Enteral formula supplemented with GOS, bifidogenic growth stimulator (BGS) and pasteurized fermented milk products (*n* = 12) vs. control enteral formula (*n* = 12).	14 w	At week 4 (A/H1N1, A/H3N2, and B)	The *Bifidobacterium* count in the test group was significantly higher (*p* < 0.05) than the control group in week 8, 12, and 18. The antibody titers against B antigen was significantly lower (*p* < 0.05) in the test group than in control.	No significant difference in other immune parameters between groups.
Bunout et al. (2002) [51]	Exploratory, randomized, blind, placebo-controlled trial	43 healthy elderly aged 70 years and older	Prebiotic (70% raftilose and 30% raftiline/6 g/day mixture; *n* = 20) vs. placebo (*n* = 23)	28 w	At week 2 (A/H1N1, A/H3N2, and B)	Significantly increased anti-Influenza B antibody titer in both prebiotic (*p* < 0.01) and placebo group (*p* < 0.01) compared to baseline.	No significant difference in other immune parameters between groups.

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
