# Peer review of "Exploring the Effect of Probiotics, Prebiotics, and Postbiotics in Strengthening Immune Activity in the Elderly"

_vaccines, 2021, doi:10.3390/vaccines9020136_

Round 1

Reviewer 1 Report

The review “Effectiveness of probiotics, prebiotics, and postbiotics in strengthening immune activity in the elderly" by Hiroyasu Akatsu does not make much sense, although it is a reasonably comprehensive summary of published papers in this area. The overarching reasons are: (i) the effect of X-biotics (where X = pro, pre or post) has never been conclusively proven; (ii) many, if not all, studies start with the biased hope of finding a positive effect, even though several have found none; (iii) the author belongs in this group and cites essentially all his own papers, which unfortunately also did not provide any conclusive beneficial effect of the Xbiotics.

Strangely, after providing largely negative or inconclusive data, the author writes a positive Conclusion. This apparent contrast is found throughout the review, after every section, some examples of which are as follows, with Line #s.

121-123: Although the immune-response-promoting functions of these beneficial microbes and their byproducts have been widely reported, their effectiveness as adjuvants in vaccination, especially for the elderly, remains unconfirmed. Only limited studies with small sample sizes are available…

12-13 (p. 5, new line numbers started from this page, following the Table): Unfortunately, no differences were detected in the subjects’ antibody titers.

37-41: Taken together, these clinical findings support the notion that the prolonged ingestion of BB536 37 is a potential prophylactic approach to improving the innate immunity of elderly patients and acts in an integrated manner to boost the effectiveness of influenza vaccines and to reduce the incidence of influenza and fever in elderly subjects. However, further large-scale clinical trials are required to confirm this definitively.

45-46: However, only limited studies of postbiotics have shown that they enhance the efficacy of vaccines in the elderly.

51-53: Unfortunately, no differences were detected in the serum Ig levels (IgA, IgM, and IgG), NK cell activities, or bactericidal or phagocytic activities of neutrophils between the MoLac and placebo groups throughout the study.

56-57: Although the ingestion of MoLac-1 tended to improve the effects of these vaccines in the elderly, the small sample size limited the accuracy and precision of the outcomes.

63-64: Consistent with the previous study, no significant differences in immune parameters were observed…

71-72: However, further large-scale studies are required to fully understand the effects of postbiotics in promoting the immune functions of this group.

78-80: These findings suggest that prebiotics boost the abundances of Lactobacillus and Bifidobacterium in the gut, with immunoregulatory effects on the host. However, only limited studies have addressed the immunoregulatory effects of prebiotics in the elderly [45,56–59].

93-96; However, the bias in the serum nutritional indices, such as the total protein and albumin levels, in Group F led to uncertainty in the ability of the prebiotics to modulate vaccine efficacy, the immune responses, and the GM in this intervention.

106-107: Because the number of studies is limited, the effects of prebiotics in enhancing immune functions and vaccination efficacy are still inconclusive.

            Inexplicably, the lack of evidence has not deterred the author in making positive conclusions at the same time, which is also all over the manuscript; such contradictions are everywhere, as exemplified below.

Abstract: This review summarizes the use of these gut modulators as adjuvants to boost both the innate and acquired immune responses in the elderly under nutritional control. Clinical trials have demonstrated that these gut modulators tend to improve the immune responses of the elderly to vaccination against influenza. However, further large-scale studies are required to fully understand the effectiveness of these approaches.

Table 1 is full of such contradictions.

106-109 (the first sentence was cited before): Because the number of studies is limited, the effects of prebiotics in enhancing immune functions and vaccination efficacy are still inconclusive. However, past studies have indicated that the administration of prebiotics tends to maintain the antibody titers against influenza antigens in elderly subjects for an extended period.

112-121: Conclusion. The whole conclusion is a mix of contradictions, and as a result, the accompanying Fig. 2 appears fictitious and hypothetical, in which the three Xbiotics have simply been added to the cellular pathways of immunomodulatory network.

The main point is: What is the pressing need to make a positive-effect statement when the scientific data are lacking or unsupportive of any effect??

After reading this review, an unbiased reader would feel that it is time to quit the Xbiotic research, especially the ones that are far-fetched, going from the general gut microbiome to improving a specific vaccine efficacy. They are technologically advanced, but so far have come up empty. Sadly, to this day, the best application of probiotics is its use to replenish the intestinal flora after an antibiotic treatment to restore bacterial metabolites and bowel movement. But even those can be achieved by ingestion of plain yogurt with active cultures, which is much more natural, frugal and tasty than the expensive commercial capsules of billions of the essentially the same bacteria that are also sold as an unregulated "dietary supplement", not a FDA-approved "pharmaceutical", bypassing any regulatory oversight. This field really has not produced much useful observation beyond the folklore and beyond Metchnikoff's original insightful suggestion about fermented milk products and good health more than a century ago!

Additional detailed comments are listed below.

1) Intriguingly, the reviews is replete with Dr. Akatsu's own publications, sometimes covering a whole paragraph, even when they were small-scale or inclusive by the author's own honest admission; examples: line 4 (p. 5), 47, 82; Table 1 is a glaring example, in which 4 out of 6 of the cited papers are Dr. Akatsu's own. At least, six references are from his group and/or collaborations: 30, 49, 54, 56, 57. It is not necessarily "inappropriate", but the danger in such cases is that the mutually self-supporting studies may offer the misleading impression of a tangible evidence, although it was not replicated by another group.

2) In cases where a small effect was seen in the long term, the study was conducted over several weeks, such as 14 weeks. What is its practical use, when seasonal influenza is an annual event? In other words, when annual vaccination is recommended, running one BB536 course for several months to find an arguably minor effect is not very sensible.

3) Describe the "placebo" or the "control group" in detail. This is important. Hopefully, it is an irrelevant but safe bacteria of equal number (in the order of billions), and not just water.

Author Response

January 29, 2021

Prof. Ralph A. Tripp

Editor-in-Chief

Vaccines

Dear Dr. Tripp,

On behalf of all authors, I am resubmitting our revised manuscript entitled “Effectiveness of probiotics, prebiotics, and postbiotics in strengthening immune activity in the elderly” (vaccines-1059237).

               We sincerely appreciate the constructive comments and points raised by you and the reviewers. We have carefully considered all comments and suggestions and have revised our manuscript in accordance with each of these points. The revised manuscript text is marked with light blue highlighting, changed part in response to the reviewer`s comment and yellow highlighting, changed part after proofreading in English. References have also been added, so their numbers are changed. Your comments have enabled us to substantially improve our manuscript. We hope that you will find our revised manuscript suitable for publication in Vaccines.                We have provided our point-by-point responses to the comments of the reviewers below. We thank you for your kind consideration of this submission. 

Sincerely yours,

Hiroyasu Akatsu, M.D., Ph.D.

Department of Community-Based Medicine, Nagoya City University Graduate School of Medicine 1 Aza Kawasumi, Mizuho-cho, Mizuho-ku, Nagoya City, Aichi, 467-8601, Japan

Tel: +81-52-853-8537

Point-by-point response to Reviewer #3 s’ comments

Reply: Thank you very much for your time and valuable feedback. We have carefully considered your comments and suggestions and have revised our manuscript accordingly. We have included our responses to the reviewers’ comments below. For clarity, the reviewer’s comments are shown in blue, and our responses are shown in black.

The revised manuscript text is marked with light blue highlighting, changed part in response to the reviewer`s comment and yellow highlighting, changed part after proofreading in English. References have also been added, so their numbers are changed.

Reviewer reports:

Reviewer #3:

The review “Effectiveness of probiotics, prebiotics, and postbiotics in strengthening immune activity in the elderly" by Hiroyasu Akatsu does not make much sense, although it is a reasonably comprehensive summary of published papers in this area. The overarching reasons are: (i) the effect of X-biotics (where X = pro, pre or post) has never been conclusively proven; (ii) many, if not all, studies start with the biased hope of finding a positive effect, even though several have found none; (iii) the author belongs in this group and cites essentially all his own papers, which unfortunately also did not provide any conclusive beneficial effect of the Xbiotics.

Reply: We thank the reviewer for taking their time and effort to evaluate our manuscript. We appreciate your valuable time, feedback, and candid opinion.

We agree that although some of the findings demonstrated a tendency to improve the effectiveness of influenza vaccination in the elderly, there were several limitations including nutritional bias, sample size, and strain specificity. The effects of X-biotics on the efficacy of vaccination in the elderly cannot be conclusively proven at present, but that is why our manuscript aimed to collect the available information and point out the limitations of these studies, providing the basis for future studies to confirm the effects of X-biotics on vaccination.

Strangely, after providing largely negative or inconclusive data, the author writes a positive Conclusion. This apparent contrast is found throughout the review, after every section, some examples of which are as follows, with Line #s.

Reply: Regarding the points you highlighted in your comments, I have corrected the relevant parts, as detailed below.

121-123: Although the immune-response-promoting functions of these beneficial microbes and their byproducts have been widely reported, their effectiveness as adjuvants in vaccination, especially for the elderly, remains unconfirmed. Only limited studies with small sample sizes are available…

Reply: Thank you for bringing this to my attention. You are correct. To address this issue, I have added the following sentence to Lines #124-6, p3 with reference #55 as “Studies have demonstrated that they affect the immune system by raising mucin levels and promoting the development of claudin and occludin, confirming their potential as possible functional ingredients [55].” and the legend for Figure 2 (Line #274-5, p9): “These points highlight the potential of an adjuvant effect but await confirmation by future research.”

12-13 (p. 5, new line numbers started from this page, following the Table): Unfortunately, no differences were detected in the subjects’ antibody titers.

Reply: Thank you for this point. I have added specific numerical values as follows on Lines #147–51, p6 “Unfortunately, no intergroup differences were detected in the subjects’ antibody titers. The ingestion of BB536 significantly improved the anti-H1N1 antibody titers (week 6, P < 0.05) in elderly patients, to antibody titers ≥ 20. The ingestion of BB536 also tended to increase the serum IgA levels (week 6, P = 0.09; week 16, P = 0.07) compared with those in the placebo group.”

37-41: Taken together, these clinical findings support the notion that the prolonged ingestion of BB536 is a potential prophylactic approach to improving the innate immunity of elderly patients and acts in an integrated manner to boost the effectiveness of influenza vaccines and to reduce the incidence of influenza and fever in elderly subjects. However, further large-scale clinical trials are required to confirm this definitively.

Reply: Thank you for this point. You are correct. I have changed the wording of this statement as follows on Lines #174–5, p6: However, clinical data on its effects on acquired immunity are still limited and await future large-scale studies.”

45-46: However, only limited studies of postbiotics have shown that they enhance the efficacy of vaccines in the elderly.

51-53: Unfortunately, no differences were detected in the serum Ig levels (IgA, IgM, and IgG), NK cell activities, or bactericidal or phagocytic activities of neutrophils between the MoLac and placebo groups throughout the study.

56-57: Although the ingestion of MoLac-1 tended to improve the effects of these vaccines in the elderly, the small sample size limited the accuracy and precision of the outcomes.

Reply: Thank you for this point. You are right. I have changed the wording as follows on Lines #191–3, p7: “Although the ingestion of L. paracasei MoLac-1 tended to improve the effects of these vaccines in the elderly, no effect was observed on other immunological parameters. The small sample size limited the accuracy and precision of the study outcomes.”

63-64: Consistent with the previous study, no significant differences in immune parameters were observed…

71-72: However, further large-scale studies are required to fully understand the effects of postbiotics in promoting the immune functions of this group.

Reply: Regarding this point, the original manuscript described work by Maruyama et al. (2016) on Lines #201–5, p7: “Interestingly, the antibody responses to type A/H1N1 and B antigens were significantly improved (P < 0.05) in the oldest old subgroup (aged ≥ 85 years; n = 11) of the LP group. This outcome suggests that heat-killed L. paracasei MCC1849 enhanced the immune functions of the oldest old subjects, who suffered the greatest immunosenescence.”

78-80: These findings suggest that prebiotics boost the abundances of Lactobacillus and Bifidobacterium in the gut, with immunoregulatory effects on the host. However, only limited studies have addressed the immunoregulatory effects of prebiotics in the elderly [45,56–59].

93-96; However, the bias in the serum nutritional indices, such as the total protein and albumin levels, in Group F led to uncertainty in the ability of the prebiotics to modulate vaccine efficacy, the immune responses, and the GM in this intervention.

106-107: Because the number of studies is limited, the effects of prebiotics in enhancing immune functions and vaccination efficacy are still inconclusive.

            Inexplicably, the lack of evidence has not deterred the author in making positive conclusions at the same time, which is also all over the manuscript; such contradictions are everywhere, as exemplified below.

Reply: Thank you for your frank opinion. I take the above points seriously and have made changes to the text to address this issue, as detailed in the responses below.

Abstract: This review summarizes the use of these gut modulators as adjuvants to boost both the innate and acquired immune responses in the elderly under nutritional control. Clinical trials have demonstrated that these gut modulators tend to improve the immune responses of the elderly to vaccination against influenza. However, further large-scale studies are required to fully understand the effectiveness of these approaches.

Reply: Thank you for your comment. I have made changes to the abstract description in the last sentences on Lines #18–22, p1: “Although the clinical evidence on this topic is limited and the initial findings await clarification through future studies with large sample sizes and proper study designs, they highlight the necessity for additional research in this field, especially in light of the ongoing COVID-19 pandemic, which is disproportionately affecting the elderly.”

Table 1 is full of such contradictions.

Reply: Table 1 is summary of previous studies whose work is addressed in the present review. The numbers of cases and p-values from these studies have been added to the table.

106-109 (the first sentence was cited before): Because the number of studies is limited, the effects of prebiotics in enhancing immune functions and vaccination efficacy are still inconclusive. However, past studies have indicated that the administration of prebiotics tends to maintain the antibody titers against influenza antigens in elderly subjects for an extended period.

Reply: Thank you for your comment. I have changed the description on Lines #245–8, p8 as follows: “These studies imply that the administration of prebiotics may maintain the antibody titers against influenza antigens in elderly subjects for an extended period; however, because the number of studies is limited, the effects of prebiotics on enhancing immune functions and vaccination efficacy are still inconclusive.”

112-121: Conclusion. The whole conclusion is a mix of contradictions, and as a result, the accompanying Fig. 2 appears fictitious and hypothetical, in which the three Xbiotics have simply been added to the cellular pathways of immunomodulatory network.

Reply: Thank you for your comment. I acknowledge that the scientific evidence is still inadequate, but I believe that this is the only way to describe it in the current situation where intervention studies for the elderly have not been sufficiently conducted. We derived our hypotheses, illustrated in Figure 2, from the findings of previous papers, and we cannot find any scientific studies that refute these ideas.

The main point is: What is the pressing need to make a positive-effect statement when the scientific data are lacking or unsupportive of any effect??

Reply: Thank you for your very meaningful question.

I think the question of how to enhance immunity and vaccine efficacy in the elderly, whose vaccine efficacy is weak, is an urgent issue. This has already been mentioned in the introduction.
Past studies have demonstrated the immunomodulatory effects of X-biotics, and some of the studies mentioned in this review show a positive trend for influenza vaccination. However, as we said throughout the review, we cannot draw a definite positive conclusion based on these small-scale studies. For these reasons, we make positive-effect statements, while concurrently urging for larger-scale clinical studies to be conducted in the future.

After reading this review, an unbiased reader would feel that it is time to quit the Xbiotic research, especially the ones that are far-fetched, going from the general gut microbiome to improving a specific vaccine efficacy. They are technologically advanced, but so far have come up empty. Sadly, to this day, the best application of probiotics is its use to replenish the intestinal flora after an antibiotic treatment to restore bacterial metabolites and bowel movement. But even those can be achieved by ingestion of plain yogurt with active cultures, which is much more natural, frugal and tasty than the expensive commercial capsules of billions of the essentially the same bacteria that are also sold as an unregulated "dietary supplement", not a FDA-approved "pharmaceutical", bypassing any regulatory oversight. This field really has not produced much useful observation beyond the folklore and beyond Metchnikoff's original insightful suggestion about fermented milk products and good health more than a century ago!

Reply: I agree with your valuable opinion. However, there seems to be a misunderstanding of our research. Although some of our studies used dietary supplements (probiotic powder), other studies used jelly and yogurt (Akatsu et al. (2013)[57], Maruyama et al. (2016) [58], Akatsu et al. (2016) [59], Nagafuchi et al. (2013)[60]). The regulation of food in Europe and the United States may be different from that in Japan, but in Japan all of these products are considered to be foods not supplements.

Additional detailed comments are listed below.

1) Intriguingly, the reviews is replete with Dr. Akatsu's own publications, sometimes covering a whole paragraph, even when they were small-scale or inclusive by the author's own honest admission; examples: line 4 (p. 5), 47, 82; Table 1 is a glaring example, in which 4 out of 6 of the cited papers are Dr. Akatsu's own. At least, six references are from his group and/or collaborations: 30, 49, 54, 56, 57. It is not necessarily "inappropriate", but the danger in such cases is that the mutually self-supporting studies may offer the misleading impression of a tangible evidence, although it was not replicated by another group.

Reply: Thank you for this point. I agree with you. Indeed, this review focuses on our research. We did a literature search on X-biotics intervention studies relating to influenza vaccination of the elderly, and there are no other relevant studies. A systematic review by Yeh, T.-L et al. (ref #32, Lines #361, p10) has analyzed all papers about the effects of X-biotics on influenza vaccines in great detail; notably, within their review, they also cite all of our studies in the section focusing on the elderly population.

2) In cases where a small effect was seen in the long term, the study was conducted over several weeks, such as 14 weeks. What is its practical use, when seasonal influenza is an annual event? In other words, when annual vaccination is recommended, running one BB536 course for several months to find an arguably minor effect is not very sensible.

Reply: Because these items are foods, it is quite normal to consume them on a daily basis. BB536 is available as a conventional yogurt in Japan. Almost all elderly people in Japan are vaccinated against influenza every year, but unfortunately, epidemics have still occurred, and many elderly people have died from secondary infections, such as pneumonia. The effectiveness of the vaccine clearly needs to be further improved. Therefore, there are needs to both develop more effective vaccines and increase the effectiveness of vaccines through lifestyle modification. In addition, we believe that X-biotics with a vaccination enhancement effect may also have an anti-infection effect, such as alleviating the symptoms in cases where an infection does occur.

3) Describe the "placebo" or the "control group" in detail. This is important. Hopefully, it is an irrelevant but safe bacteria of equal number (in the order of billions), and not just water.

Reply: The content of each “placebo” or “control group” has now been clearly described throughout the manuscript (Lines #140, 162, p6; Lines #197–8, 220–5, p7; and Lines #236–7, p8).

Reviewer 2 Report

This manuscript entitled “Effectiveness of probiotics, prebiotics, and postbiotics in strengthening immune activity in the elderly” presents an interesting review where the author has summarized the use of these gut modulators as adjuvants to boost the innate as well as acquired immune responses in the elderly under nutritional control. The strength of the review is that the content is quite informative and well supported by with self-explanatory figures and observations from the recent available literature. Though the modulation of gut environment with use of use of probiotics, prebiotics, or postbiotics has been shown to improve the immune responses and the trends are promising, more extensive studies are required to establish it conclusively. The authors have also highlighted this fact in the present study. Overall the data has been well presented well and would be of interest to the readers. Before forwarding the manuscript for acceptance, I would like the authors to address a couple of minor issues

  1. Figure1: Since the trend is projection or extrapolation, the figure legend should be modified to “predicted global population trend by age groups…. “.
  2. Any specific reason to limit the data to 2007

Author Response

January 29, 2021

Prof. Ralph A. Tripp

Editor-in-Chief

Vaccines

Dear Dr. Tripp,

On behalf of all authors, I am resubmitting our revised manuscript entitled “Effectiveness of probiotics, prebiotics, and postbiotics in strengthening immune activity in the elderly” (vaccines-1059237).

               We sincerely appreciate the constructive comments and points raised by you and the reviewers. We have carefully considered all comments and suggestions and have revised our manuscript in accordance with each of these points. The revised manuscript text is marked with light blue highlighting, changed part in response to the reviewer`s comment and yellow highlighting, changed part after proofreading in English. References have also been added, so their numbers are changed. Your comments have enabled us to substantially improve our manuscript. We hope that you will find our revised manuscript suitable for publication in Vaccines.                We have provided our point-by-point responses to the comments of the reviewers below. We thank you for your kind consideration of this submission. 

Sincerely yours,

Hiroyasu Akatsu, M.D., Ph.D.

Department of Community-Based Medicine, Nagoya City University Graduate School of Medicine 1 Aza Kawasumi, Mizuho-cho, Mizuho-ku, Nagoya City, Aichi, 467-8601, Japan

Tel: +81-52-853-8537

Point-by-point response to Reviewer #4 s’ comments

Reply: Thank you very much for your time and valuable feedback. We have carefully considered your comments and suggestions and have revised our manuscript accordingly. We have included our responses to the reviewers’ comments below. For clarity, the reviewer’s comments are shown in blue, and our responses are shown in black.

The revised manuscript text is marked with light blue highlighting, changed part in response to the reviewer`s comment and yellow highlighting, changed part after proofreading in English. References have also been added, so their numbers are changed. 

Reviewer reports:

Reviewer #4:

This manuscript entitled “Effectiveness of probiotics, prebiotics, and postbiotics in strengthening immune activity in the elderly” presents an interesting review where the author has summarized the use of these gut modulators as adjuvants to boost the innate as well as acquired immune responses in the elderly under nutritional control. The strength of the review is that the content is quite informative and well supported by with self-explanatory figures and observations from the recent available literature. Though the modulation of gut environment with use of use of probiotics, prebiotics, or postbiotics has been shown to improve the immune responses and the trends are promising, more extensive studies are required to establish it conclusively. The authors have also highlighted this fact in the present study. Overall the data has been well presented well and would be of interest to the readers. Before forwarding the manuscript for acceptance, I would like the authors to address a couple of minor issues

Reply: Thank you very much for your comments.

  1. Figure1: Since the trend is projection or extrapolation, the figure legend should be modified to “predicted global population trend by age groups…. “.

Reply: The figure legend for Figure 1 has been amended as suggested in Lines #64–8, p2.

  1. Any specific reason to limit the data to 2007.

Reply: The chart in Figure 1b was generated based on the data from U.S. Department of Health and Human Service, Centers for Disease Control and Prevention, cited as reference #12* in Lines #310-2, p9.

Beyond the year 2010, all available data are presented on a yearly basis (for example, 2011–2012 or 2012–2013)

* Thompson, M.G.; Shay, D.K.; Zhou, H.; Bridges, C.B.; Cheng, P.Y.; Burns, E.; Bresee, J.S.; Cox, N.J. Estimates of deaths associated with seasonal influenza-United States, 1976-2007. Morb. Mortal. Wkly. Rep. 2010, 59, 1057–1062. (in reference #12 in Lines #310-2, p9)

Reviewer 3 Report

Journal:

Vaccines

Author:

Hiroyasu Akatsu

Manuscript title:

Effectiveness of probiotics, prebiotics, and postbiotics in strengthening immune activity in the

elderly

Dear Sirs:

I found the manuscript “Effectiveness of probiotics, prebiotics, and postbiotics in strengthening immune activity in the elderly” by Hiroyasu Akatsu, interesting, correct and within the scope of the journal; however, I found it also not publishable in its actual format.

First of all, the manuscript is a “review”. I'm not specialist in the field, and you have to consider this, but searching through the internet I found at least two additional reviews of similar subjects that are not cited in the manuscript (Zimmermann and Curtis 2017, Praharaj et al. 2015). I recommend the author have a close look to both manuscripts, and include these and other key references in his study.

Second of all, the statistical approach is too loose. You have to cite which was the test used (i.e. Student, ANOVA...), the number of patients and, finally, the probability (level of significance). From my point of view, saying that this or that treatment is or not significant without all these details is not enough.

If these two main points are not adequately addressed, I don't consider the quality of the manuscript is acceptable enough to be published.

Zimmermann P and Curtis N. The influence of probiotics on vaccione responses- A systematic review. http://dx.doi.org/10.1016/j.vaccine.2017.08.069

Praharaj I, John SM, Bandyopadhyay R, Kang G. Probiotics, antibiotics and the immune responses to vaccines. http://dx.doi.org/10.1098/rstb.2014.0144

Author Response

January 29, 2021

Prof. Ralph A. Tripp

Editor-in-Chief

Vaccines

Dear Dr. Tripp,

On behalf of all authors, I am resubmitting our revised manuscript entitled “Effectiveness of probiotics, prebiotics, and postbiotics in strengthening immune activity in the elderly” (vaccines-1059237).

               We sincerely appreciate the constructive comments and points raised by you and the reviewers. We have carefully considered all comments and suggestions and have revised our manuscript in accordance with each of these points. The revised manuscript text is marked with light blue highlighting, changed part in response to the reviewer`s comment and yellow highlighting, changed part after proofreading in English. References have also been added, so their numbers are changed. Your comments have enabled us to substantially improve our manuscript. We hope that you will find our revised manuscript suitable for publication in Vaccines.                We have provided our point-by-point responses to the comments of the reviewers below. We thank you for your kind consideration of this submission. 

Sincerely yours,

Hiroyasu Akatsu, M.D., Ph.D.

Department of Community-Based Medicine, Nagoya City University Graduate School of Medicine 1 Aza Kawasumi, Mizuho-cho, Mizuho-ku, Nagoya City, Aichi, 467-8601, Japan

Tel: +81-52-853-8537

Point-by-point response to Reviewer #5 s’ comments

Reply: Thank you very much for your time and valuable feedback. We have carefully considered your comments and suggestions and have revised our manuscript accordingly. We have included our responses to the reviewers’ comments below. For clarity, the reviewer’s comments are shown in blue, and our responses are shown in black.

The revised manuscript text is marked with light blue highlighting, changed part in response to the reviewer`s comment and yellow highlighting, changed part after proofreading in English. References have also been added, so their numbers are changed. 

Reviewer reports:

Reviewer #5:

I found the manuscript “Effectiveness of probiotics, prebiotics, and postbiotics in strengthening immune activity in the elderly” by Hiroyasu Akatsu, interesting, correct and within the scope of the journal; however, I found it also not publishable in its actual format.

Reply: Thank you very much for your kind comments and pointing out the significance of our work.

  1. First of all, the manuscript is a “review”. I'm not specialist in the field, and you have to consider this, but searching through the internet I found at least two additional reviews of similar subjects that are not cited in the manuscript (Zimmermann and Curtis 2017, Praharaj et al. 2015). I recommend the author have a close look to both manuscripts, and include these and other key references in his study.

Reply: Citations to these two reviews have been added as recommended on Line #102-3, p3 and Line #364–7, p10 as references #33–4. Additionally, citations to several other relevant studies have also been added to the revised manuscript on Line #108, p3 and Lines #384–92, p11 as references #41–4. The reference numbering in our manuscript was updated accordingly.

  1. Second of all, the statistical approach is too loose. You have to cite which was the test used (i.e. Student, ANOVA...), the number of patients and, finally, the probability (level of significance). From my point of view, saying that this or that treatment is or not significant without all these details is not enough.

Reply: The number of patients and the level of significance for each study has been added throughout the manuscript and in Table 1. In particular, the significance of differences between three groups is described on Lines #239–44, p8. However, the type of test used was not included because this is not a systemic review, and it is uncommon to include such information in a standard review.

Round 2

Reviewer 1 Report

The author has responded to all my comments. However, the title is somewhat misleading as the "Effectiveness" was not conclusively proven. I suggest to add "An analysis of the" in front. In the Table, 4 out 6 references are still the author's own; I wish papers from other groups were also be cited. It is unclear if the author did not cite or did not find them, since no "point by point response" was submitted for each comment.

Author Response

Point-by-point response to Reviewer #1 s’ comments

Reply: Thank you very much for your time and valuable feedback. We have carefully considered your comments and suggestions and have revised our manuscript accordingly. We have included our responses to the reviewers’ comments below. For clarity, the reviewer’s comments are shown in blue, and our responses are shown in black.

The revised manuscript text is marked with light blue highlighting, changed part in response to the reviewer`s comment. References have also been added, so their numbers are changed. 

Reviewer reports:

Reviewer #1:

The author has responded to all my comments. However, the title is somewhat misleading as the "Effectiveness" was not conclusively proven. I suggest to add "An analysis of the" in front.

Reply:
In according to your suggestion, the title has been amended as “Exploring the effect of probiotics, prebiotics, and postbiotics in strengthening immune activity in the elderly.”

In the Table, 4 out 6 references are still the author's own; I wish papers from other groups were also be cited. It is unclear if the author did not cite or did not find them, since no "point by point response" was submitted for each comment.

Reply:
Additional studies from other groups have been included in Table 1 (p5-6) and description in the text as follow and the several reference orders was changed after reference #57.

FYI, only studies involved elderly was added into Table.

Additional description in the text

Line#135, p4: In this review, only some of the studies will be discussed.

Line#186-188, p7: In addition, some studies have been performed on the immune activity of elderly by Lactobacillus strains [56, 57]. Bosch et al. (2012) demonstrated that 3-month supplementation of Lactobacillus plantarum CECT 7315/7316 significantly improved influenza-specific IgA level [58].

Line #279-282, p9: Similar to the above observations, Bunout et al. (2002) [51] reported that administration of a prebiotic mixture (70% raftilose + 30% raftiline) or placebo significantly increased anti-Influenza B antibody titer in both prebiotic (P < 0.01) and placebo group (P < 0.01) compared to baseline, but without inter-group difference.

In addition, due to the review by myself, there was an error in the number of people in the second row from the bottom in Table 1, so I corrected it as follows.

“Enteral formula supplemented with GOS, bifidogenic growth stimulator (BGS) and pasteurized fermented milk products (n =12) vs control enteral formula (n = 11).”

Reviewer 3 Report

Dear Sirs:

All the changes suggested were added. I find that the manuscript has improved a lot and, although I'm not a specialist in vaccines, I consider that the work can be interesting if it will be published.

Yours sincerely,

Author Response

Point-by-point response to Reviewer #3 s’ comments

Reviewer reports:

Reviewer #3:

Dear Sirs:

All the changes suggested were added. I find that the manuscript has improved a lot and, although I'm not a specialist in vaccines, I consider that the work can be interesting if it will be published.

Yours sincerely,

Reply: Thank you very much for your kind comments and pointing out the significance of our work.

In addition, due to the review by myself, there was an error in the number of people in the second row from the bottom in the table, so I corrected it as follows.

“Enteral formula supplemented with GOS, bifidogenic growth stimulator (BGS) and pasteurized fermented milk products (n =12) vs control enteral formula (n = 11).”

By reviewer #1 offer, the studies from other groups have been included in Table (p5-6) and description in the text (Line #135, p4, Line #186-188, p7 and Line #279-282, p9).

So, the several reference orders was changed after reference #57.

FYI, only studies involved elderly was added into Table.
